# Chemometric Analysis-Based Sustainable Use of Different Current Baking Wheat Lots from Romania and Hungary

Adrian Vasile Timar [ID], Alin Cristian Teusdea *, Cornelia Purcarea [ID], Adrian Nicolae Vuscan *, Adriana Ramona Memete [ID] and Simona Ioana Vicas [ID]

Department of Food Engineering, Faculty of Environmental Protection, University of Oradea, 410048 Oradea, Romania; atimar@uoradea.ro (A.V.T.); cpurcarea@uoradea.ro (C.P.); memeteadriana25@gmail.com (A.R.M.); svicas@uoradea.ro (S.I.V.)
* Correspondence: ateusdea@uoradea.ro (A.C.T.); avuscan@uoradea.ro (A.N.V.)

**Abstract:** Wheat is the most important raw material for bakery industries. Real-time grain quality assessment could increase bakery product quality and baking efficiency. The quality assessment of wheat grains can be conducted using modern and non-invasive techniques based on near-infrared spectrophotometry (NIRS) methods for the assessment of gluten content (WetGL), protein content, Zeleny index (ZelenyIdx), grain humidity (Ur), etc. The topic covered in the study is of current interest, is a part of sustainable research, and involves aspects of food quality, one of the concerns addressed by the University of Oradea's Department of Food Engineering. The present study was carried out in 2020 on eleven wheat lots from Romania and Hungary. Following the NIRS analyses, the results show varied quality for the Romanian and Hungarians wheat lots. The Romanian variety Crisana recorded the highest values for quality parameters, being similar to the Hungarian variety Bekes from Hajdu Bihar County. The statistical analysis was carried out using multivariate analysis (multivariate analysis of variance (MANOVA), canonical variate analysis (CVA) and hierarchical cluster analysis (HCA)), which highlighted which of the batches of wheat grains can be mixed to obtain a raw material of high quality for the bakery industry.

**Keywords:** wheat grains; near-infrared spectrophotometry; quality chemical analysis; multivariate analysis of variance





## 1. Introduction

Wheat cropping is a longtime activity of humankind. Baking wheat milling factories transform the grain into flour, which is then sold to bread-making facilities. The assessment of wheat grain quality started to be conducted using modern and non-invasive techniques in order to reduce the time and increase the precision of wheat evaluation. The application of modern non-destructive testing methods on grain quality testing had important implications for grain production, distribution and sale. Using modern non-destructive testing methods to assess the grain quality can reduce costs and improve the detection accuracy [1–4]. The new approach in wheat grain quality control is based on NIRS methods. Those methods are recommended because of their precision, reliability, repeatability, cost effectiveness, and superior data management and especially because they are not time consuming [5,6]. There are several studies that have demonstrated that NIRS is widely used in determining the quality and safety of agricultural products, such as the assessment of aflatoxin and fungal contamination or the identification of pest insect species found in various grains [2,5–8].

This assessment equipment is very reliable, can operate in severe environments like harvesting plots in the field, warehouses, mills and even cars or containers. That allows the assessment of the wheat grains from the beginning and provides good batch quality for delivery.

The application of modern non-destructive testing methods on grain quality testing could reduce costs and improve detection accuracy; furthermore, when combined with modern computer technology and data processing, it would play an important role in agriculture. Compared with traditional methods of chemical analysis, the main technical characteristics of near-infrared spectroscopy (NIRS) are fast analysis, simultaneous determination of multi-component samples without pretreatment, non-destructive analysis, distance measurement and real-time analysis, low analysis cost and simple operation. The prospective uses of NIRS are very promising, and it could be applied in the rapid detection and online detection of hazardous substances, other constituents of the grains (even pests) and nutrients combined with the new instrument and chemical analysis methods developed or under development. In this way, compared with classical methods, the new NIRS equipment is superior [2,5,6,8].

Another application of FT-NIR (Fourier transform near-infrared) and FT-MIR (mid-infrared) spectroscopy in combination with chemometric methods was developed by De Girolamo et al. (2018) to identify and classify a large volume of wheat bran contaminated with DON (Deoxynivalenol) [9].

Data resulting from the measurements are safely recorded on internal storage drive, memory stick or other storage devices and can be delivered via the internet to specific locations. The most important advantages of this method are the short measuring time, nondestructive analysis and the very good ratio cost/effectiveness. An operator with average training can conduct the measurement in less than two minutes [5,6,8].

Chemical analysis used as chemometric analysis is also a statistical method, but in contrast with classical statistics, it is focused on the extraction of functional information among variables [10,11]. In recent decades, chemometry techniques have been intensively applied to the simultaneous determination of compounds in a specified sample [10,12–15].

Process analytical technology instruments are used in the real-time inspection of a process line in all food sectors. They include measurement tools, process–product analyzers and sensors. As process analytical technology tools developed from univariate measuring methods (temperature, pH, color and pressure) to multivariate measuring systems, it was determined that this system might provide information regarding the physical, chemical and biological properties of the materials being investigated [16].

Also, European regulations were laid down considering these parameters as key parameters and there are specifically noticed levels of these parameters and actions related to deviations from these values, decreased prices, restrictions in public intervention schemes from an acquisitions point of view, export banning, etc. The values for the studied parameters laid down by Regulation (EU) no. 1272/2009 of the Commission from 11 December 2009 establishing the rules for implementing Regulation (EC) no. 1234/2007 of the Council regarding the purchase and sale of agricultural products within the public intervention scheme and Romanian Standard SR 13548/2013 are the following: Organoleptical properties—specifically for healthy products according to variety and lots description, hectolitre mass (Mh)—minim 75 kg [17,18], 1000 grains mass (M1000) between 40–60 g, MS between 1.19–1.28 g/cm$^3$ [19,20], protein content—10.5%, WetGL—22%, Ur—14.5%, ZelenyIdx—minim 22 mL [17,18].

Chemometrics is applied to solve both descriptive and predictive problems in experimental natural sciences, especially in chemistry [14,21]. In descriptive applications, properties of chemical systems are modeled with the intent of learning the underlying relationships and structure of the system. In predictive applications, properties of chemical systems are modeled with the intent of predicting new properties or behavior [10,21].

Developing tools for evaluating the quality of the raw material in order to generate high quality products is especially essential for the bakery industry, where tons of products are produced every day [16].

Our main goal in this work was to show that it is possible to predict and rank grain quality using physical and NIRS information (protein content, WetGL, Ur and Zeleny Idx) as well as modern statistical analysis. Another objective was to demonstrate how the

results of the chemometric analysis might be used to determine which wheat lots should be combined to produce a high-quality product with excellent baking capabilities. This work is novel considering that it integrates non-invasive analytical techniques with multivariate analysis to acquire relevant information for the estimation of products with applications in the bakery sector.

## 2. Materials and Methods

### 2.1. The Biological Material

The present study was carried out in 2020, using both wheat lots imported to Romania from Hajdu Bihar County, Hungary, and wheat lots grown in Bihor County, Romania. Eleven wheat batches from Romania and Hungary were obtained for study (Table 1). The number of repetitions was 8 for each batch taken into the study.

**Table 1.** Wheat lots' sample coding.

| No. | Sample Code | Sample Location |
|---|---|---|
| 1. | DROPIA | Romania, Săcuieni |
| 2. | ALEX_LOT1 | Romania, Săcuieni |
| 3. | ALEX_LOT2 | Romania, Săcuieni |
| 4. | ALEX_LOT3 | Romania, Săcuieni |
| 5. | CRIŞANA | Romania, Săcuieni |
| 6. | BEKES | Hungary, Berettyóújfalu |
| 7. | MV_KOLO | Hungary, Berettyóújfalu |
| 8. | MV_VEKNI | Hungary, Berettyóújfalu |
| 9. | MV_MARSHALL | Hungary, Berettyóújfalu |
| 10. | MV_EMESE | Hungary, Berettyóújfalu |
| 11. | MV_MAGDALENA | Hungary, Berettyóújfalu |

For better data management, the samples were coded. The sample codes for each wheat lot are listed in Table 1.

### 2.2. Methods

Following the harvest, the grain lots were cleaned with a winnower and grader and then stored for a short time (5 days) before analysis. The samples were analyzed the same day after being received in the laboratory. Preparation of samples was conducted by removing foreign parts in accordance with European legislation [17,18,22].

The quality parameters evaluated in our study were Mh, M1000, MS, Protein content, WetGL, Ur and ZelenyIdx. The parameters were chosen because of their relevance for flour quality [7,23]. Wheat lots were collected and analyzed in eight replicates for each wheat lot ($n = 8$). The methods used for analysis are in accordance with Romanian standards and are cited from latest scientific studies [21,24]. Sampling was conducted with cylindrical probes from the surface to the bottom layers.

#### 2.2.1. Organoleptic Analysis

Organoleptic analysis was conducted just in order to select the proper samples. The organoleptic test was carried out using 5 testers who are members of the University of Oradea's Food Engineering Department who are taking part in the "Sensory Analysis of Food Products" course and have competence in this field of study. For the evaluation of wheat lots, the panel approach was used in terms of smell, taste, color and appearance. The samples were graded, and acceptance was granted for samples that met the minimum criteria according to each lot (Romanian Standard SR 13548:2013, ISO 7970:2021—Wheat (*Triticum aestivum* L.)—Specification and Romanian National Standards for Wheat STAS 6253-80) [22,25,26]. The ISO 7970 standard specifies minimum requirements for wheat (*Triticum aestivum* L.) grains intended for human consumption and traded internationally. It is also applicable to the local wheat trade. Impurities, damaged wheat grains, broken

grains, low-value wheat, milled grains, immature grain, black point grain, grain attacked by pests and other elements are detailed in the site https://www.iso.org/standard/75731.html (accessed on 13 December 2022) [22]. The scoring was conducted in panels with grades from 1 to 10. The lowest score was 1, and the highest was 10. The acceptance value for each parameter was 7.

The samples were classified as accepted, and the samples that met the minimum criteria according to each variety and the Romanian Standard SR 13548:2013, ISO 7970:2021—Wheat (*Triticum aestivum* L.)—Specifications and Romanian National Standards for Wheat STAS 6253-80 [17,18] were studied.

### 2.2.2. Chemical Analysis

For the chemical parameters, an NIRS AgriCheck spectrometer from Bruins InstrumentsTM (KPM Analytics Technology Drive, Westborough, MA 01581) was used. This instrument uses near-infrared light to evaluate the sample spectra and determine the composition of the sample. The wavelength range used was from 730 mn to 1100 nm with a wavelength data increment of 0.5 nm. The AgriCheck is an NIRS cost-effective solution for grain and seed analysis. This approach is similar to other recent studies that have studied applications of NIRS to the analysis of wheat, oilseed and rice components, pests and even contaminants [27]. The chemical and rheological parameters determination was conducted using whole grains as samples. The samples were cleaned, and all foreign parts and damaged grains were removed. Then, samples were placed in the feeder hopper, and the following measurements were made: ZelenyIdx (mL), WetGL (%), Protein content (%) and UR (%). For statistical analysis, data were exported from the database to the computer.

### 2.2.3. Physical Analysis

Granomat™ (PFEUFFER GMBH, Flugplatzstraße 70 97318 Kitzingen, Germany) and Acculab™ analytical scales (4802 Glenwood Rd. Brooklyn, NY 11234, USA) were used to determine the physical parameters, which also provide very good precision, reliability and repeatability. Data were processed with an Acer Aspire™ 5733 laptop (Acer's EMEA headquarters, Lugano, Switzerland). The physical parameters determination was conducted using whole grains as samples. The samples were cleaned, and all foreign parts and damaged grains were removed. After that, they were weighed (600 g) and put in the feeder hopper, and there were measured the temperature, weight and reflection [23,24,28].

The M1000 (1000 grains mass) was measured using the Acculab™ analytical scale; for M1000, the 1000 grains were counted in 5 repetitions and weighed.

The MS (specific mass) was determined using a pycnometer and 2 g of grains weighed with the Acculab™ analytical scale that were immersed in toluene.

The values were calculated using following Equation (1) [24]:

$$\gamma = \frac{m_1}{m_1 + m_2 + m_3}\rho \quad \text{g}/cm^3 \tag{1}$$

$m_1$—mass of the weighed sample, g;
$m_2$—mass of pycnometer with liquid, g;
$m_3$—mass of pycnometer with sample and liquid, g;
$\rho$—MS of the toluene used, g/cm$^3$.

### 2.3. Statistical Analysis

Each parameter/variable was subjected to one-way ANOVA ($p$ = 0.05), and furthermore to post hoc multiple pairwise sample means comparisons using Tukey's test. This test generates comparison results as letters accompanying the mean values (Table 2). Different letters along the columns denote statistically significant differences ($p$ = 0.05) between wheat lots. The multivariate analysis included principal component analysis (PCA), multivariate analysis of variance (MANOVA, $p$ = 0.05), canonical variates analysis (CVA) and hierarchical cluster analysis (HCA) with complete linkage and Euclidean distance options

performed with the P.A.S.T. version 3.05 statistical package. Canonical variates analysis (CVA) biplot and Canonical variates analysis (CVA) 3D graphical representationwere generated with MATLAB Software v2022b.3.

**Table 2.** One-way ANOVA ($p$ = 0.05) results of analyzed wheat parameters.

| Wheat Lots | Mh (kg/hL) | M1000 (g) | MS (g/cm³) | WetGL (%) | Protein (%) | Ur (%) | ZelenyIdx (mL) |
|---|---|---|---|---|---|---|---|
| DROPIA | 77.49 a,b ± 0.04 | 47.00 c ± 0.00 | 1.23 c,d ± 0.00 | 25.11 f ± 0.21 | 13.69 a ± 0.05 | 12.29 e ± 0.05 | 62.88 f ± 1.89 |
| ALEX_LOT1 | 77.89 a ± 0.04 | 47.36 a,b,c ± 0.29 | 1.25 b,c ± 0.03 | 25.11 f ± 0.30 | 13.42 b,c,d,e ± 0.04 | 12.23 e ± 0.06 | 75.25 a ± 0.71 |
| ALEX_LOT2 | 78.16 a ± 0.05 | 47.24 b,c ± 0.04 | 1.11 h ± 0.05 | 24.89 f ± 0.09 | 12.99 g ± 0.19 | 12.81 b ± 0.05 | 73.13 a,b ± 1.13 |
| ALEX_LOT3 | 76.96 b,c ± 0.29 | 47.49 a,b ± 0.20 | 1.15 f,g,h ± 0.02 | 25.34 d,e ± 0.00 | 13.58 a,b,c ± 0.12 | 12.26 e ± 0.05 | 71.50 b,c ± 1.31 |
| CRISANA | 76.36 c ± 1.21 | 47.60 a,b ± 0.00 | 1.19 d,e,f ± 0.02 | 26.01 c ± 0.13 | 13.23 e,f ± 0.02 | 12.49 d ± 0.11 | 68.38 c,d,e ± 1.85 |
| BEKES | 77.58 a,b ± 0.19 | 47.24 b,c ± 0.04 | 1.20 d,e ± 0.01 | 26.79 b ± 0.15 | 13.60 a,b ± 0.08 | 12.57 c,d ± 0.06 | 70.50 b,c ± 2.33 |
| MV KOLO | 78.09 a ± 0.14 | 47.76 a ± 0.07 | 1.17 e,f,g ± 0.01 | 26.07 c ± 0.03 | 13.41 b,c,d,e ± 0.07 | 12.33 e ± 0.10 | 68.88 c,d ± 2.10 |
| MV VEKNI | 77.73 a ± 0.15 | 47.26 b,c ± 0.04 | 1.30 a ± 0.04 | 25.36 d ± 0.09 | 13.52 a,b,c,d ± 0.19 | 12.22 e ± 0.05 | 62.00 f ± 1.77 |
| MV MARSHALL | 78.09 a ± 0.15 | 47.05 c ± 0.54 | 1.29 a,b ± 0.02 | 25.12 e,f ± 0.08 | 13.38 c,d,e,f ± 0.07 | 13.01 a ± 0.08 | 65.00 e,f ± 4.2 |
| MV EMESE | 77.48 a,b ± 0.46 | 47.62 a,b ± 0.46 | 1.23 c,d ± 0.01 | 27.15 a ± 0.07 | 13.36 d,e,f ± 0.12 | 12.77 b ± 0.14 | 65.50 d,e,f ± 3.02 |
| MV MAGDALENA | 78.05 a ± 0.15 | 47.55 a,b ± 0.22 | 1.14 g,h ± 0.01 | 26.12 c ± 0.09 | 13.19 f,g ± 0.20 | 12.63 c ± 0.06 | 62.13 f ± 0.83 |

Results are expressed as the mean ± SD ($n$ = 8). Different letters presented in superscript denote statistically significant differences ($p$ = 0.05) between wheat lots.

The proposed multivariate sequence was used to get the proper number of clusters. In this way, we know which clusters maximize which parameters' values, and so we can prescribe which wheat lot is suitable for different bakery and bread products. Also, these results can provide information about how certain wheat lots can be mixed from different clusters to provide a specific or desired flour quality. Furthermore, the MANOVA uses the canonical axes of the samples to calculate the multiple pairwise comparisons in a multivariate way. This is another the reason we considered the CVA in conjunction with MANOVA. The HCA was used because its results corelate very well with the CVA and MANOVA clustering results and provide the dissimilarity distance threshold that generates the CVA and MANOVA clusters. Additionally, the clustering evolution with respect to the dissimilarity distance can be followed on the HCA dendrogram. This information can also provide the "similarity" between the clusters, information that is needed when wheat lots are mixed.

## 3. Results and Discussion

The parameters taken into account in this study were the following: Organoleptical properties, Mh, M1000, MS, Protein content, WetGL, Ur and ZelenyIdx. Their significance is related to processing technological requirements [21,27–29], particularly in terms of flour component quality, and the latest studies showing that they have nutritional impact in human intake, even from a functional point of view [27,29–31]. Recent studies have confirmed the correlations between wheat parameters and the quality of bakery products [32,33].

### 3.1. Organoleptic Analysis

All samples submitted for analysis passed the organoleptic test. With this fast test, the operator may avoid examining and processing the damaged grains. All organoleptic parameters were in accordance with the requirements; the scores for all samples ranged from 9 to 10 for each of the parameters.

### 3.2. Physical, Chemical and Rheological Parameters

The results according to physical, chemical and rheological parameters of the studied wheat grains recorded during the study experiences are presented in Table 2.

Grain moisture content does not directly affect grain quality but can indirectly affect quality since grain will spoil at moisture contents above that recommended for storage (15%). The Ur was not considered a key parameter because it can be corrected easily but was taken into calculation because it affects the rest of the parameters [28,34,35]. The Ur value in wheat lots tested ranged from 12.22% to 13.01% (Table 2). The grinding is affected

by some of the parameters that are not strictly related with quality and for this reason they are also kept in calculations.

There are key parameters like protein content and ZelenyIdx present in European regulations related with wheat trading. These include price ceiling, price reductions of lots with lower than standard parameters and rejecting the acquisitions. Using these parameters from the farmers' point of view can provide them with a useful tool for meeting the highest possible criteria in their production.

According to Table 2, the wetGL content in wheat lots from Romania ranged from 25.11% to 26.01%, whereas wheat types from Hungary ranged from 25.12% to 26.72%. The DROPIA wheat lot had the highest protein content, followed by the BEKES lot, at 13.69% and 13.60%, respectively. In the case of the examined lots, the ZelenyIdx value ranged from 62% (at MV VEKNI wheat lot) to 75.25% (at ALEX LOT 1 wheat lot).

On the other hand, in this study a different approach for storage facilities operators and actors in the bakery sector was presented. Table 3 shows the result of comparing parameters related to physical parameters (M1000, MS, Ur and Mh) and chemical parameters (WetGL, Protein and ZelenyIdx) important for storage operators from storage safety and grain values and only chemical parameters (WetGL, Protein and ZelenyIdx) important for actors in the bakery sector. As a result, all supply chain actors' needs from farm to fork can be addressed.

**Table 3.** Wheat lots ranking: by Lp-norm ($p = 2.5$) between protein content (%) and ZelenyIdx content (left part of the table), by Lp-norm ($p = 2.5$) between all analyzed parameters (middle part of the table) and by Lp-norm ($p = 2.5$) between protein, WetGL content (%) and ZelenyIdx content (right part of the table). Data columns implied in Lp-norm were subjected to min-max normalization.

| Wheat Lots | Lp Norm ($p = 2.5$) | Rank | Wheat Lots | Lp Norm ($p = 2.5$) | Rank | Wheat Lots | Lp Norm ($p = 2.5$) | Rank |
|---|---|---|---|---|---|---|---|---|
| a * | | | | | b * | c * | | |
| ALEX_LOT1 | 0.840 | 1 | CRISANA | 0.707 | 1 | DROPIA | 0.793 | 1 |
| ALEX_LOT3 | 0.785 | 2 | MV KOLO | 0.697 | 2 | ALEX_LOT1 | 0.715 | 2 |
| BEKES | 0.769 | 3 | MV MARSHALL | 0.676 | 3 | MV KOLO | 0.702 | 3 |
| DROPIA | 0.758 | 4 | ALEX_LOT1 | 0.669 | 4 | BEKES | 0.672 | 4 |
| ALEX_LOT2 | 0.636 | 5 | DROPIA | 0.651 | 5 | MV VEKNI | 0.645 | 5 |
| MV VEKNI | 0.578 | 6 | ALEX_LOT3 | 0.628 | 6 | MV MARSHALL | 0.550 | 6 |
| MV KOLO | 0.564 | 7 | MV MAGDALENA | 0.617 | 7 | ALEX_LOT3 | 0.541 | 7 |
| MV MARSHALL | 0.443 | 8 | MV EMESE | 0.576 | 8 | MV MAGDALENA | 0.499 | 8 |
| MV EMESE | 0.428 | 9 | MV VEKNI | 0.558 | 9 | ALEX_LOT2 | 0.450 | 9 |
| CRISANA | 0.424 | 10 | BEKES | 0.543 | 10 | CRISANA | 0.378 | 10 |
| MV MAGDALENA | 0.215 | 11 | ALEX_LOT2 | 0.483 | 11 | MV EMESE | 0.376 | 11 |

* The columns from the table represent the following: (a) by Lp-norm between protein and ZelenyIdx content, (b) by Lp-norm between all analyzed parameters and (c) by Lp-norm between protein, WetGL and ZelenyIdx content.

Table 3 shows that the Romanian wheat lots provide a good level of production. The results are shown better in Figure 1 that reveals the CVA distribution of the parameters. The wheat lots Alex and Dropia are very well situated and show the highest values. The Hungarian lot Vekni is the same group but considering the fertilization, its potential is lower. These results confirm the adaptability of the Romanian cultivars even when there is a lack of agro technique, according to the latest studies [36]. There are three groups that are emphasized in the Figure 1.

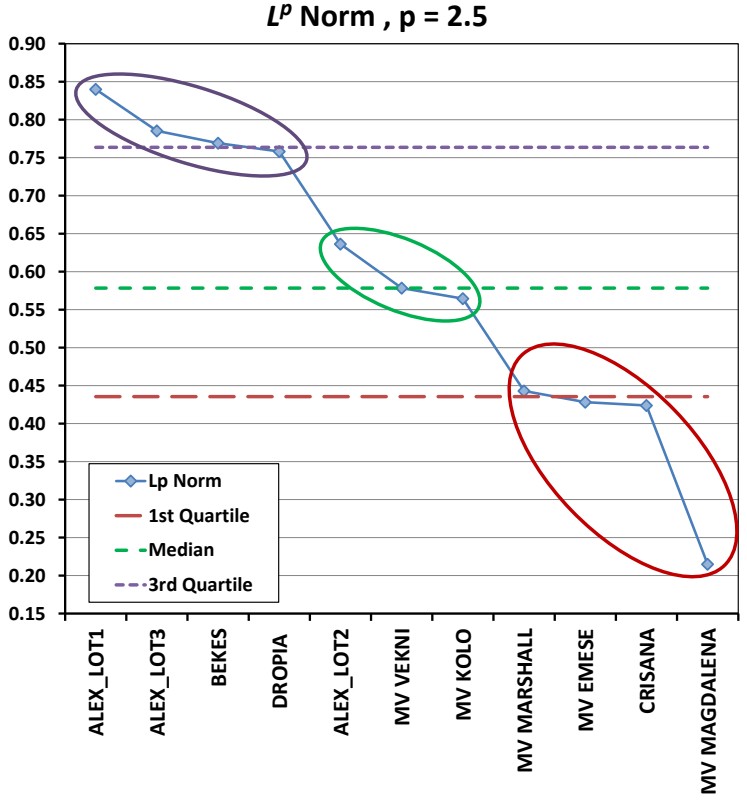

(**a**)

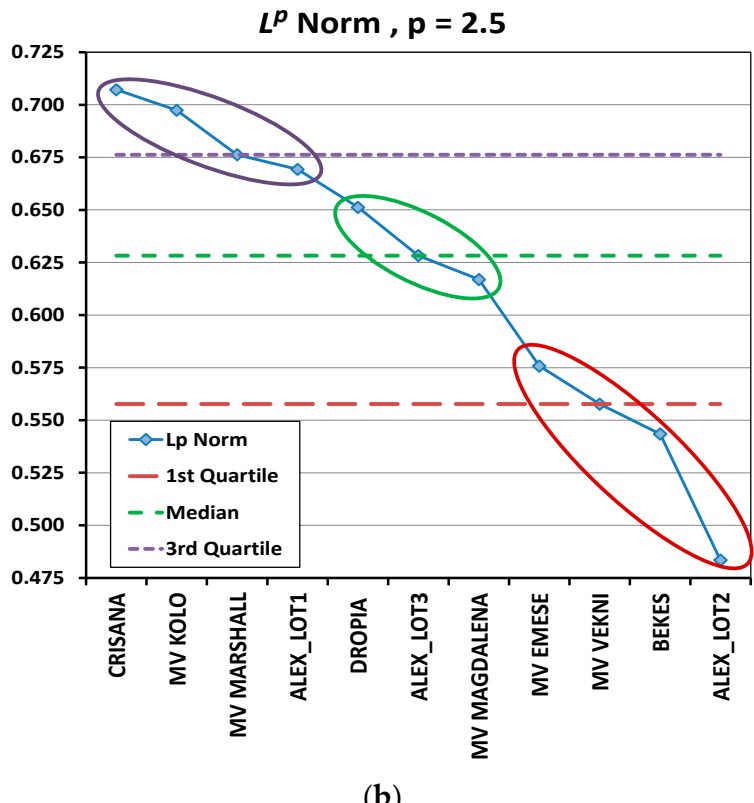

(**b**)

**Figure 1.** *Cont.*

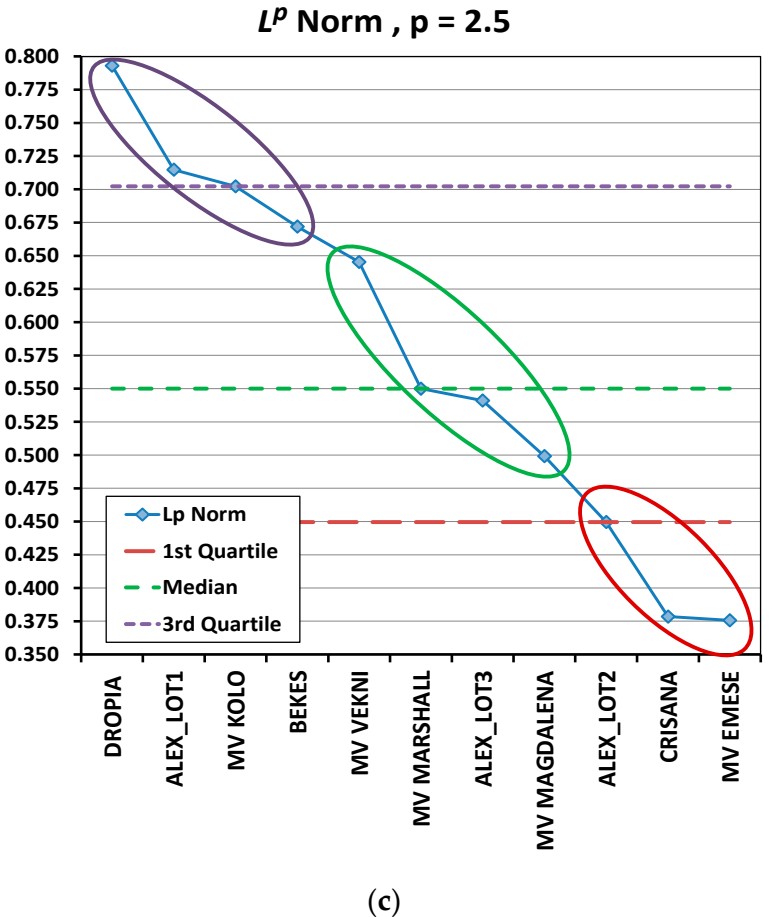

(**c**)

**Figure 1.** Graphical representation of wheat lots ranking: (**a**) by Lp-norm between protein and ZelenyIdx content, (**b**) by Lp-norm between all analyzed parameters and (**c**) by Lp-norm between protein, WetGL and ZelenyIdx content. Data columns implied in Lp-norm were subjected to min-max normalization. The ellipses gathers the lots that are corresponding to groups surrounding the 1st quartile, median and 3rd quartile (as mentioned in the legend).

### 3.3. Multivariate Analysis

Multivariate data analysis is the evaluation of data comprised of several variables collected from a variety of samples. The aim of multivariate data analysis is to identify all of the variations in the data matrix studied. As a result, chemometric tools attempt to discover correlations between samples and variables in a given data set and convert them to new latent variables [37].

The multivariate analysis was performed in order to get the clustering information for the wheat lot samples. Hierarchical cluster analysis (HCA) was performed. As HCA input, the scores of first two canonical axes of the canonical variates analysis (CVA) were considered [38–40].

The ideal composition of wheat for bakery products is based on diverse recipes, milling and production technologies and additives or even on traditions, sensory perceptions or just marketing and is just asserted theoretically.

The quality of durum wheat flour depends on the type and amount of gluten proteins and starch, while flour's nutritional value rests on metabolite contents such as polyphenols [41]. The nutrition composition of red and white wheat (from U.S. data) indicates that the intact grains are nearly identical (in case of similar cropping technologies), with only minor differences between micronutrients such as iron, zinc, phosphorus and potassium (slightly higher in white wheat), and magnesium, copper and niacin (slightly higher in red wheat) [42,43].

The generally accepted minimal requirements are provided by National and EU regulation [17,18] and due to there being no ideal proper composition for different kinds of bread or bakery products, the aim is to exceed the legally established parameters.

In some studies from the literature, multivariate analysis was performed by investigating the data with principal component analysis (PCA) [38,44]. FTIR spectroscopy combined with chemometric data processing in terms of PCA is an effective method in food-related research for the development of novel food quality indicators [45]. The initial data (i.e., measured parameters values) were standardized before PCA processing. The total explained variance of the first two components was around 50%. In this situation, the significance of the clustering process using the PCA scores would not have the appropriate statistical consistence. This was the reason for performing the canonical variates analysis (CVA) to obtain the clustering information.

The CVA, or discriminant function analysis (DFA), is based on Eigen-analysis of the multivariate data (several variables at the same time), as with the PCA, but the axes (i.e., discriminant functions) maximize the among-groups covariance matrix. Thus, this multivariate method's main advantage is that it performs the best discrimination between the groups of wheat lots. Also, the CVA is related to multivariate analysis of variance (MANOVA) [38,40]. From the MANOVA table (Table 4), the most important information is the *p*-values of the pairwise comparisons made in five dimensions (which equal the variables number). In Table 4 were highlighted the pairwise comparisons with $p > 0.05$, to emphasize the lot pairs that have no statistically significant differences.

**Table 4.** Multivariate analysis of variance (MANOVA) results among wheat samples groups with Bonferroni-corrected *p*-values.

| MANOVA *p*-Values | DROPIA | ALEX_LOT1 | ALEX_LOT2 | CRISANA | BEKES | MV_KOLO | MV_VEKNI | MV_MARSHALL | ALEX_LOT3 | MV_EMESE | MV_MAGDA |
|---|---|---|---|---|---|---|---|---|---|---|---|
| DROPIA | | 0.0164 | 0.0002 | 0.0004 | 0.0001 | 0.0011 | **0.6116** | 0.0012 | 0.0175 | 0.0000 | 0.0005 |
| ALEX_LOT1 | 0.0164 | | 0.0004 | 0.0009 | 0.0002 | 0.0053 | 0.0173 | 0.0002 | **0.1666** | 0.0000 | 0.0003 |
| ALEX_LOT2 | 0.0002 | 0.0004 | | 0.0004 | 0.0000 | 0.0002 | 0.0001 | 0.0011 | 0.0010 | 0.0000 | 0.0007 |
| CRISANA | 0.0004 | 0.0009 | 0.0004 | | 0.0207 | **0.0781** | 0.0007 | 0.0002 | 0.0086 | 0.0024 | **0.1014** |
| BEKES | 0.0001 | 0.0002 | 0.0000 | 0.0207 | | 0.0342 | 0.0002 | 0.0000 | 0.0005 | **0.0908** | 0.0081 |
| MV_KOLO | 0.0011 | 0.0053 | 0.0002 | **0.0781** | 0.0342 | | 0.0025 | 0.0001 | 0.0379 | 0.0008 | 0.0454 |
| MV_VEKNI | **0.6116** | 0.0173 | 0.0001 | 0.0007 | 0.0002 | 0.0025 | | 0.0007 | 0.0060 | 0.0000 | 0.0006 |
| MV_MARSHALL | 0.0012 | 0.0002 | 0.0011 | 0.0002 | 0.0000 | 0.0001 | 0.0007 | | 0.0002 | 0.0000 | 0.0005 |
| ALEX_LOT3 | 0.0175 | **0.1666** | 0.0010 | 0.0086 | 0.0005 | 0.0379 | 0.0060 | 0.0002 | | 0.0000 | 0.0022 |
| MV_EMESE | 0.0000 | 0.0000 | 0.0000 | 0.0024 | **0.0908** | 0.0008 | 0.0000 | 0.0000 | 0.0000 | | 0.0021 |
| MV_MAGDA | 0.0005 | 0.0003 | 0.0007 | **0.1014** | 0.0081 | 0.0454 | 0.0006 | 0.0005 | 0.0022 | 0.0021 | |

The highlighted values mark the pairwise comparisons with $p > 0.05$, to emphasize the lot pairs that have no statistically significant differences.

Figure 2 presents the CVA biplot with the first two axes (Axis 1 and Axis 2). The biplot combines the samples' CVA scores and the variables' loadings that generate the discriminant functions or axes. The total variance explained by first two axes is 76.41% and is much higher than the PCA; furthermore, the total variance explained by the first three canonical axes (Figure 3) is 91.88%. In this way, the possible clustering information generated with the scores from these two CVA axes has a high statistical significance level.

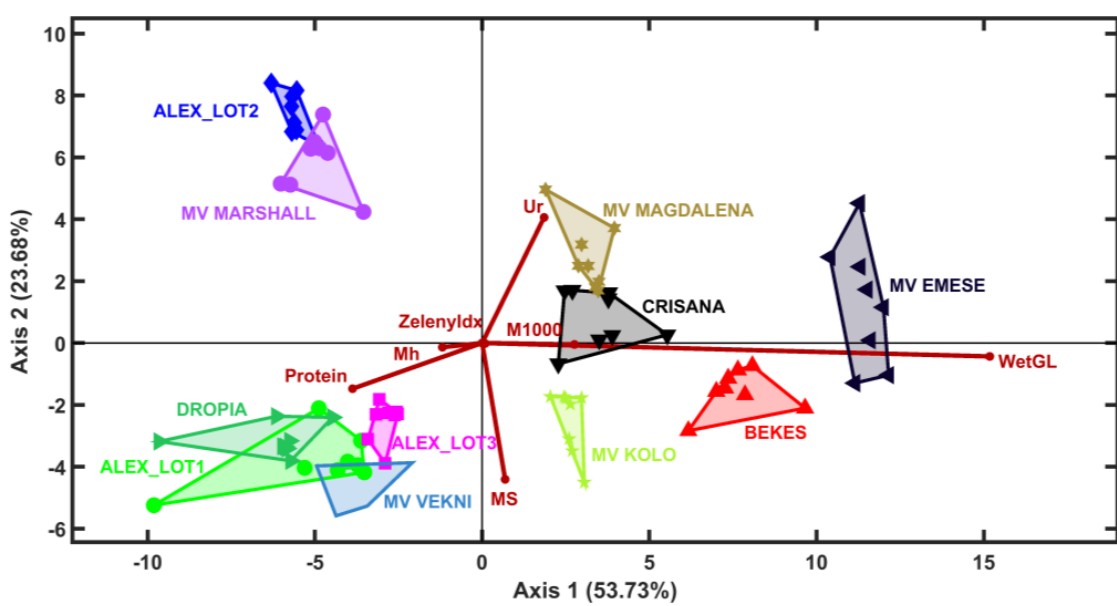

**Figure 2.** Canonical variates analysis (CVA) biplot. Sample points were represented as convex hull polygons with different color rendering. Figure was generated with MATLAB Software v2022b.3.

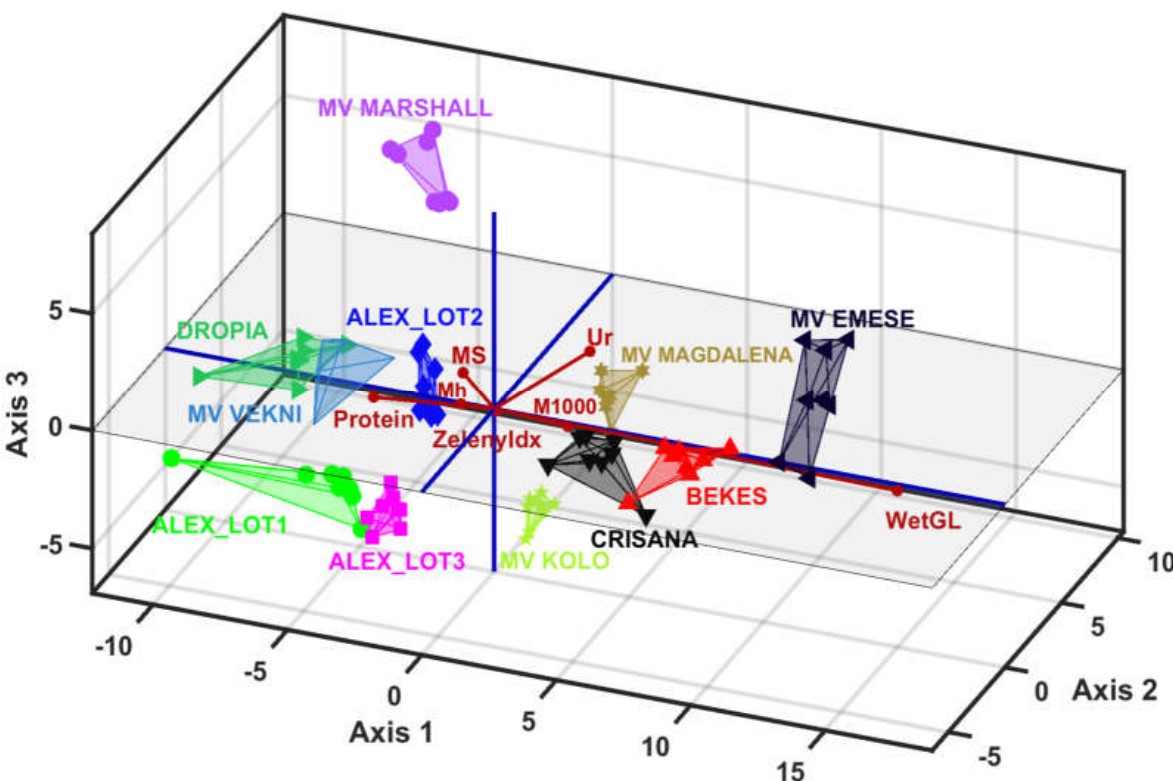

**Figure 3.** Canonical variates analysis (CVA) 3D graphical representation of samples group scores. Figure was generated with MATLAB Software v2022b.3.

The graphical representation of the variables' loadings was completed with vectors emerging from the coordinates' frame origin (Figure 2). Their end points denote the direction of the positive loading of the variable or the most abundant content direction described by the variables.

Based on that, the samples that are near a variable vector end point will display the most abundant content of that variable. On the other hand, the samples that are near the

opposed direction of a variable vector end point will display the less abundant content of that variable. In other words, the variable vectors are performing the contrast of samples' content, and thus the samples' discrimination is conducted. The CVA is generating the multivariate "contrast" or discrimination of the samples. In our experimental design, the variables' vectors have a range distribution over the four quadrants of the coordinates' frame. This reveals qualitative and quantitative differences between the wheat lots with respect to the considered variables.

The wheat samples Dropia, Alex_Lot1, Alex_Lot3, MV_Vekni, MV_Kolo and BEKES are positively loaded with the protein content and MS vectors and thus present the most abundant content of protein and specific mass. The wheat samples Dropia, Alex_Lot1, Alex_Lot3, MV_Vekni, MV_Marshall and Alex_Lot2 are positively loaded with the ZelenyIdx vector and thus present the highest level of ZelenyIdx.

The wheat samples MV_Marshall and Alex_Lot2 are positively loaded with the Mh vector and thus present the highest level of Mh. The wheat samples MV_Magdalena, Crisana, MV_Emese, MV_Marshall and Alex_Lot2 are positively loaded with the Ur vector and thus present the highest level of moisture (Ur). The wheat samples MV_Magdalena, Crisana, MV_Emese, Bekes and MV_Kolo are positively loaded with the WetGL and M1000 vectors and thus present the highest level of wet gluten (WetGL) and thousand grains mass (M1000).

The wheat samples MV_Marshall and Alex_Lot2 are positively loaded with the Mh vector and thus present the highest level of Mh. The wheat samples MV_Magdalena, Crisana, MV_Emese, MV_Marshall and Alex_Lot2 are positively loaded with the Ur vector and thus present the highest level of moisture (Ur).

The wheat samples from the first group—MV_Magdalena, Crisana, MV_Emese, Bekes and MV_Kolo—are positively loaded with the WetGL and M1000 vectors and thus present the highest level of wet gluten (WetGL) and thousand grains mass (M1000).

There are also another three groups that are visible in Figure 1. The results are also confirmed by Figure 2. The second group is formed from the Alex_Lot 2 and Marshal lots and also shows good properties regarding protein content and ZelenyIdx. The third group that is formed from Crisana, Magdalena and Kolo is not so valuable because of its lower ZelenyIdx. The fourth group formed from Emese and Bekes, despite having high WetGL, is poor in total protein content and rheological properties as shown by ZelenyIdx being lower.

Figures 2 and 3 show the mixing possibility in order to correct the quality parameters of the studied batches. In this way, the optimal combination is Group 1 and Group 3, Group 1 and Group 4, Group 2 and Group 3 and Group 1 and Group 4. Mixing group Group 1 and Group 2 is not effective because of their similar properties.

In Figures 2 and 3, the biplots combine the samples' canonical coordinates (Axis 1, Axis 2 and Axis 3) with the variable vectors. Each vector points out the highest-level content of the variables. The samples that are neighbored with the vector end point performed at a high level of that parameter. In our DOE (Design of Experiment) there are several vectors that point out some wheat samples. In this way, the sample profiles define the wheat baking quality. Furthermore, if one needs a different predefined baking quality (described by our DOE), one can mix the wheat samples based on the parameter vectors.

One of the main goals of the multivariate analysis was to assess which wheat lots have the highest content of protein and the highest values of ZelenyIdx. From the CVA biplot, it can be noticed that the wheat samples group Dropia, Alex_Lot1, Alex_Lot3 and MV_Vekni are simultaneously positively loaded with these two variables' vectors. This partially validates the results of Lp-norm ($p = 2.5$) ranking (Table 3 and Figure 1) based on the protein and ZelenyIdx variables' values.

From a legal point of view, according to the abovementioned European regulation regarding official trading of grains and intervention policies, the results shown are the following. Storage operators are focused on many parameters from a storage safety point of view, up to quality parameters that are leading to market value. In this way M1000, MS, Ur, Mh, WetGL, Protein and ZelenyIdx are key parameters for these actors. The values and

results after conducting the statistical analysis showed that CRISANA and ALEX_LOT1 from the Romanian lots and MV KOLO and MV MARSHALL from the Hungarian lots are leading due to M1000 and MS, with Ur as a regulator because of the storage costs that are strictly related to the volumes and surfaces of the storage infrastructure and the energy costs of drying for providing storage safety. Because the storage facilities of the last decades are very effective and because the processing of wheat has allowed the correction of chemical parameters, the actors from the wheat grain storage field are mostly cost oriented, and this is shown by their requirements referring to the low volume and surface demands and low water amount removed for the storage requirements. For the actors in the bakery sector, the correlation between WetGL, Protein and ZelenyIdx parameters is significant due to the technological requirements (fermentation, dough development, crumb development and nutritional values). In this way, the ranking showed that Dropia and ALEX_LOT1 from the Romanian lots and MV KOLO and BEKES are the best positioned. Data from Table 2 show that protein content is the main asset but is correlated with the gluten percentage. The ZelenyIdxat very high values are acting as a controller and can compensate for the previous slightly lower parameters' values.

In the first five places, above the Lp-norm median, the CVA results can be recognized. The single small discrepancy on MV_Vekni is due to the fact that CVA is performing the grouping for seven variables and the Lp-norm was calculated only for two variables. The HCA process also designated this wheat lot group as a valid cluster (Figure 4). The other HCA generated clusters generally agreed with the CVA grouping and MANOVA pairing. The discrepancies between the HCA and CVA and MANOVA results derive from the fact the HCA dendrogram and CVA biplot (Figure 2) are performed in two dimensions from the seven-dimensional analysis results. In contrast, the MANOVA exclusively generates results only from the seven-dimensional analyses, with no possibility of reducing the number of variables.

The hierarchical cluster analysis (HCA) dendrogram that reveals the mathematically based group structures is shown in Figure 4.

The Magdalena and Crisana lots are also a good option, especially when using drying systems that will increase the quality out of the concerns regarding contamination that can irreversibly affect the batch quality.

The CVA and HCA small discrepancies are due to the fact the CVA biplot is generated as a 2D projection on the Axis 1 and Axis 2 plane from seven dimensions of data, and HCA has the input of only the scores of Axis 1 and Axis 2 from the CVA. This can be seen in Figure 3 where the CVA scores are 3D graphically represented over Axis 1, Axis 2 and Axis 3. Here the wheat samples Alex_Lot2 and MV_Marshall are separated over Axis 3 (result validated by MANOVA, Table 4), but in HCA there are contained in one cluster.

Our results reveal that there are differences between Romanian and Hungarian wheat lots.

This confirms studies of other environmental conditions and allows quantification of the results, as the correlated statistical method used allows us to predict and emphasize the grains' quality. Also, real-time assessment of quality parameters through non-invasive and whole-grain methods based on NIRS and Granomat allow prediction of mixing batches in order to obtain the optimal grain quality in the new batches. This was based on the measurements and showed also the direct correlation between parameters.

Multivariate analysis was performed on wheat lot samples in order to assess the wheat lots that have the highest content of protein and highest level of ZelenyIdx at the same time. The canonical variates analysis (CVA) biplot emphasizes wheat lots Dropia, Alex_Lot1, Alex_Lot3 and MV_Vekni in response to this demand. Also, the hierarchical cluster analysis (HCA) and multivariate analysis of variance (MANOVA) results validate and offer consistent statistical significance accuracy for the HCA conclusion.

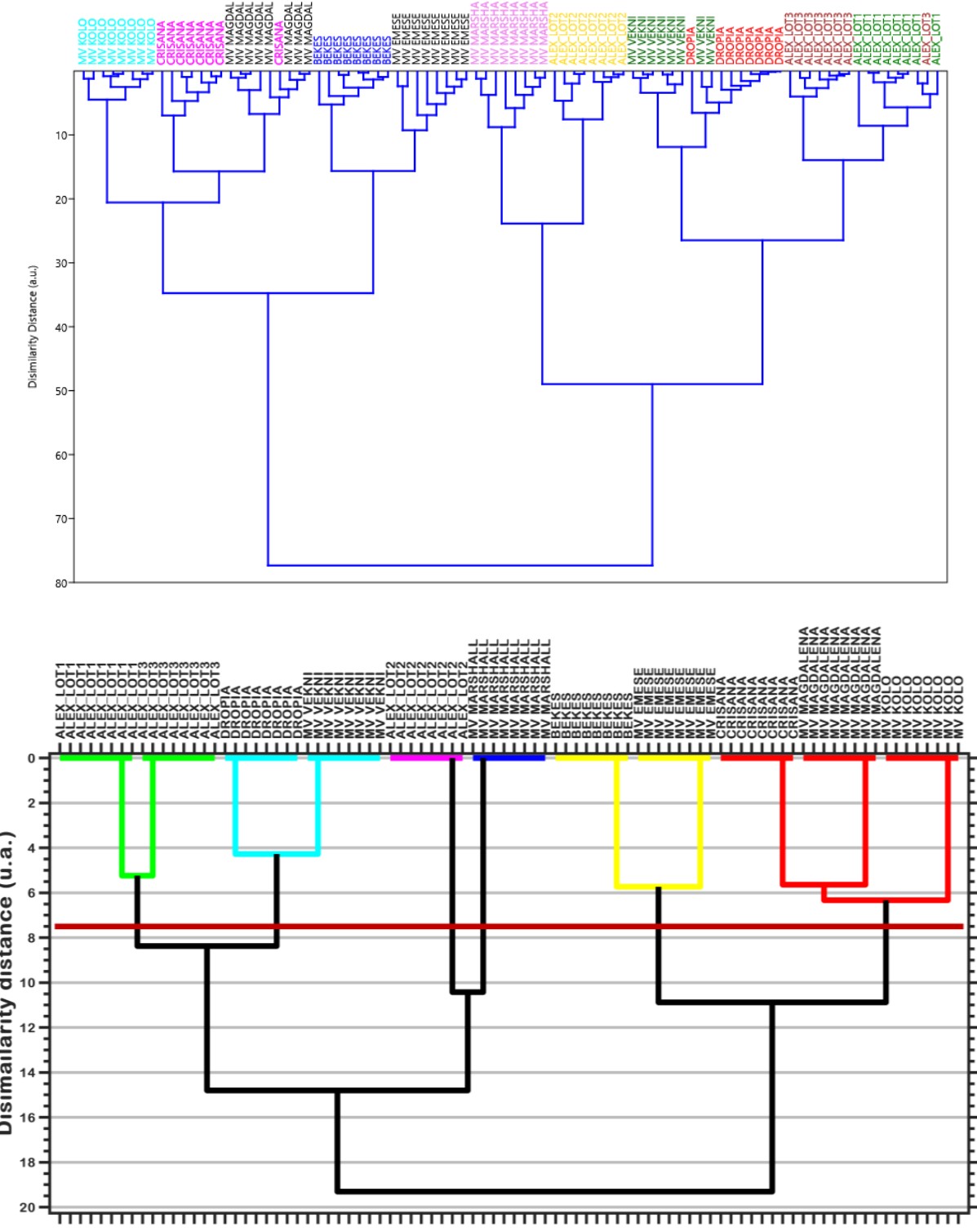

**Figure 4.** Hierarchical cluster analysis (HCA) dendrogram. The HCA method used the "complete-linkage" option to compute the results. The red colored horizontal line marks the dissimilarity distance threshold that generates the lots clusters in MANOVA (*p* = 0.05) results. Each lots cluster is marked, also, with different color.

## 4. Conclusions

Multivariate analysis was performed on all wheat samples in order to assess the wheat lots that have the highest content of protein and highest value of ZelenyIdx at the same time. The canonical variates analysis (CVA) biplot emphasizes that wheat lots

Dropia, Alex_Lot1, Alex_Lot3 and MV_Vekni were in the same cluster and are suitable for use in mixing batches as correctors. Also, the hierarchical cluster analysis (HCA) and multivariate analysis of variance (MANOVA) results validate and offer consistent statistical significance accuracy for the HCA conclusion. The statistical methods used allows us to predict and emphasize the quality of grain batches in real time. Also, those methods allow the predicting of batch mixing in order to obtain the optimal grain quality in new batches. This was based on the measurements and showed also the direct correlation between parameters. Finally, the obtained results demonstrate that the statistical methods used allow predicting and highlighting the quality of the grains. With this cross-validation approach, other relevant parameters specific to each batch can be introduced, such as pest presence or various contaminants.

**Author Contributions:** Conceptualization, A.V.T., S.I.V. and A.N.V.; methodology, A.C.T., C.P., A.N.V. and A.V.T.; formal analysis, A.C.T. and A.V.T.; investigation, A.C.T., C.P., A.N.V., A.R.M. and A.V.T.; writing—original draft preparation, A.C.T. and A.V.T.; writing—review and editing, A.V.T., A.R.M. and S.I.V.; visualization, A.V.T.; supervision, A.V.T. and S.I.V.; funding acquisition, A.N.V. and A.V.T. All authors have read and agreed to the published version of the manuscript.

**Funding:** Publishing of this research was funded by University of Oradea.

**Institutional Review Board Statement:** Not applicable.

**Informed Consent Statement:** Not applicable.

**Data Availability Statement:** Not applicable.

**Acknowledgments:** The research has been supported through the grant competition "Scientific Week of the University of Oradea", project number 121/06.25.2021.

**Conflicts of Interest:** The authors declare no conflict of interest.

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
