# Peer review of "Chemometric Analysis-Based Sustainable Use of Different Current Baking Wheat Lots from Romania and Hungary"

_sustainability, doi:10.3390/su151712756_

Round 1
Reviewer 1 Report
The use of chemiometrics in food analysis should be better introduced, by uderling the applications in simple foods, processed foods and food preparations and related references added such as:
De Girolamo, et al. Fourier transform near-infrared and mid-infrared spectroscopy as efficient tools for rapid screening of deoxynivalenol contamination in wheat bran. J. Sci. Food Agric. 2018.
Durazzo, et al. Qualitative Analysis of Traditional Italian Dishes: FTIR Approach. Sustainability 2018, 10, 4112. https://doi.org/10.3390/su10114112
N. Kumar, A. Bansal, G.S. Sarma, R.K. Rawal Chemometrics tools used in analytical chemistry: An overview. Talanta, 123 (2014), pp. 186-199
Results in Table 2 should be better described in the text.
Graphical representation of wheat cultivars ranking should be better described in the text.
Figure 2 should be better discussed in the text.
Moderate english revision is requested.
Reviewer 2 Report
8 reps were collected, was there any power analysis to determine this or is it just a number that the authors have selected? Since this is a biological sample, sampling plan should've been drafted prior to the study.
Why 5 testers? And how & how long were they trained for this study?
Can the authors expand a bit more on what is ISO7970 to provide context for the readers?
Any reasons why Tukey's was used but not Fisher's?
The intention of running MANOVA, CVA, and HCA is needed. More information is also needed for the multivariate analysis, did the authors assume similar equality of covariance for example in CVA? Or were there any class corrections applied? For HCA, what clustering did u use? Did u normalise the data? More information is needed
Table 3. What's Lp-norm and p = 2.5? Need to be expanded or introduced in the manuscript. It seems that there were some degree of normalisation? More info needed in the stats section.
HCA as no results? Why only the first 2 scores of the canonical axis? How many % did it explained? More info needed.
Table 4. What's in the cell? The F value? I don't quite understand what's happening here. How did the author correct for Bonferroni? It looks like p = 0.6116 is highlighted red meaning that it's sig.?
Please park all the results in results section, how come I'm seeing MANOVA and CVA in Discussion?
Fig 2. How did the author draw these areas? Wouldn't a centroid approach with confidence ellipses work better?
These justificaitons are required before I review the results further.
English reads fine
Reviewer 3 Report
In this paper, the authors suggest that it is possible to predict and rank grain quality using physical and NIRS information as well as modern statistical analysis. Another objective was to demonstrate how the results of the chemometric analysis might be used to determine which wheat lots should be combined to produce a high-quality product with excellent baking capabilities. This paper it is interesting considering that it integrates non-invasive analytical techniques with multivariate analysis to acquire relevant information for the estimation of products with applications in the bakery sector.
Some of the information presented in this manuscript could be useful in related fields, if the work is done methodically correctly.
In my opinion, the manuscript contains interesting results and appropriate conclusions. The study's methodology can be useful for future research and development in the field. After carefully manuscript reading, I think, that presented experiment is valuable. However, I have listed some suggestions for correction that the authors must consider.
First, it is necessary to mention the absence of information on the main agrochemical parameters of the soil as well as about the climatic conditions in the areas of samples origin. The authors should not use their own pronoun; "we". Please check for the entire manuscript.
The Discussion section could provide a more in-depth interpretation of the results and relate them to previous studies or practical implications.
L. 408-417. This paragraph is suitable for the Introduction section, in no case for the Conclusions. So I suggest you to summarize only the clear conclusions derived from the results obtained.
L. 509, 563, 573. Write the scientific name in italics. Please check the entire manuscript.
L. 558. Please complete reference no. 33 because it seems to me incomplete in this form. Please check the entire Reference section.
Round 2
Reviewer 2 Report
OK clear for the response.
I'm unsure if the Editors are OK with this, any biological sampling would require sufficient statistical power to be estimated prior to publication. Selecting repetition of 8 just because there are 7 response variables is not a legitimate reason.
Sensory analysis with 5 members of the faculty is concerning, usually most sensory analysis is done with at least 8 trained panel. The author also still did not include any information on the competence that the members have. Did they receive training? How many hours? Reference materials?
I understand that the combo of MANOVA, CVA, and HCA is used to 'provide the best decision' for classification. But more details is needed. Why did the author selected specifically CVA for example, why not other multidimensional technique say a PCA or GPA? Same story with MANOVA or HCA (rather than k-means?)
So was there any assumptions of equality of covariance? The author did not answer this.
For HCA, ok complete linkage is used so please add that information for the agglomeration method, and what distance did the author use for clustering? Still of infos are missing.
The authors did not answer how many % variance explained by the canonical axes.
Table 4. If in red are p > 0.05, so why is it on the last column I see 0.0081 not in red?
English reads fine
Round 3
Reviewer 2 Report
The authors have justified this and please ensure you add the information from the previous review to the manuscript.
If the Editor is OK with the sensory & statistical justification, then I'm fine.